# Immune Response and Outcome of High-Risk Neuroblastoma Patients Immunized with Anti-Idiotypic Antibody Ganglidiomab: Results from Compassionate-Use Treatments

**DOI:** 10.3390/cancers14235802

**Published:** 2022-11-25

**Authors:** Leah Klingel, Nikolai Siebert, Sascha Troschke-Meurer, Maxi Zumpe, Karoline Ehlert, Stefanie Huber, Hans Loibner, Oliver Mutschlechner, Holger N. Lode

**Affiliations:** 1Pediatric Hematology and Oncology, University Medicine Greifswald, 17475 Greifswald, Germany; 2Anyxis Immuno-Oncology GmbH, 1230 Vienna, Austria

**Keywords:** neuroblastoma, immunotherapy, ganglioside GD2, vaccine

## Abstract

**Simple Summary:**

The treatment of high-risk neuroblastoma patients with anti-GD2 antibodies has improved survival, and it is an established treatment strategy; however, many patients still experience a late relapse. One disadvantage of passive immunotherapy is the absence of a memory response. Therefore, developing an active immunotherapy leading to a sustained immune response may provide a solution and prevent the occurrence of late relapses following anti-GD2 antibody therapy. Here, we describe the first-in-man compassionate use of the ganglidiomab vaccine following passive immunotherapy with an anti-GD2 antibody (dinutuximab beta) in seven neuroblastoma patients. The vaccine was well-tolerated, and all patients not pre-treated by haploidentical transplantation developed vaccine-specific immune responses.

**Abstract:**

(1) Background: High-risk neuroblastoma (HR-NB) is associated with a poor prognosis despite a multimodal high-intensity treatment regimen, including immunotherapy with anti-GD2 monoclonal antibodies (mAb). Here, we investigated the effects of an anti-idiotypic vaccine based on the mAb ganglidiomab that structurally mimics GD2. (2) Methods: Patients with HR-NB treated with anti-GD2 mAb dinutuximab beta and who achieved complete remission after frontline or salvage therapy were offered the vaccine (0.5 mg ganglidiomab adsorbed to Alhydrogel^®^). Side effects (CTCAE v4.03) and immune responses were determined on each visit. We also evaluated the time to relapse or progression until the last follow-up. (3) Results: Seven HR-NB patients (five frontlines, two relapsed) received 6–22 subcutaneous injections every two weeks. Six of the seven patients showed an immune response. The non-responding patient had a haploidentical stem cell transplantation as part of the previous treatment. No fever, pain, neuropathy, or toxicities ≥ grade 3 occurred during or post-treatment. All immunized patients did not experience relapses or progressions of their neuroblastoma. (4) Conclusions: This is the first-in-man use of the ganglidiomab vaccine, which was well-tolerated, and all patients not pre-treated by haploidentical transplantation developed vaccine-specific immune responses. These findings provide an important basis for the design of prospective clinical trials.

## 1. Introduction

Neuroblastoma (NB) is the most common extracranial, solid tumor of childhood and is a largely heterogeneous disease entity that can be stratified into low, intermediate, and high-risk groups. Approximately half of all patients are classified as high-risk (HR) at the time of diagnosis [1]. Treatment of HR-NB patients consists of high-intensity chemotherapy, surgery, myeloablative therapy with autologous stem cell rescue, radiation therapy, and treatment with Iodine-131 metaiodobenzyl guanidine (MIBG) [2]. Despite progress in treatment outcomes over the past years [3,4], the event-free survival (EFS) rate at five years for patients undergoing a multimodal treatment regimen reaches only about 50% [5,6,7]. Therefore, the development of new therapeutic strategies, such as immunotherapies, is imperative. 

The first clinically effective immunotherapy for NB is based on passive immunotherapy using mAbs against the disialoganglioside (GD2). GD2 is a suitable target as it is abundantly expressed on tumor cells (up to 10^7^ molecules per cell [8]), with a limited expression on normal tissue, and forms a stable complex once bound to an Ab [9]. Moreover, loss of the GD2 antigen from tumors rarely happens as an escape mechanism after receiving Ab therapy [10].

Anti-GD2 mAb can activate human leukocytes to mediate Ab-dependent cell-mediated cytotoxicity (ADCC) and monocyte-macrophage-mediated phagocytosis [3]. When the Fc receptors on granulocytes and natural killer (NK) cells are activated by the mAb attached to the tumor cells, cytotoxic granules and cytokines are released, thereby killing the tumor cell through ADCC. Furthermore, these Fc-receptors also induce phagocytosis by stimulating monocytes and macrophages through the Ab-dependent cellular phagocytosis (ADCP) mechanism [11]. Although NB patients receiving high-dose chemotherapy show an attenuated lymphocyte response, neutrophils and macrophages are only temporarily deactivated. Hence, ADCC and ADCP, which depend on these effector cells, can still mediate tumor cell eradication. In addition to ADCC, complement-dependent cytotoxicity (CDC) is another essential effector function of mAbs. When complement C1q attaches to the complement binding region of a tumor-bound mAb, it starts a complement activation cascade prompting the generation of a membrane attack complex (MAC), leading to the formation of pores in the cell membrane and the lysis of tumor cells. [3,9]. 

Clinical findings in the past decades with murine and chimeric mAbs have demonstrated activity and efficacy in treating HR-NB. However, because of the clearance of injected mAbs over time, the use of active immunotherapy, i.e., a vaccine which leads to the continuous production of an anti-GD2 Ab in the immunized patient, is a growing alternative. Such cancer vaccines have been named the “next generation” strategy to overcome the short-lived immune response seen after administering mAbs. In this context, a bivalent vaccine is currently being evaluated in clinical trials that target two antigens bound on NB cells: GD2L and GD3L [12]. In one of the trials, an induction of persistent anti-GD2 and anti-GD3 Ab responses was observed, and a high anti-GD2 IgG1 titer was found to be independently correlated with favorable results. Furthermore, the progression-free survival (PFS) was 32 ± 6%, and the overall survival was 71 ± 7% at five years [13].

Therefore, developing a vaccine that can provide a sustained anti-tumor response for immunized patients with prolonged protection from relapse may be advantageous. 

Here, we investigated the immune response of a ganglidiomab vaccine in seven patients with HR-NB in compassionate-use treatments. We hypothesized that patients with HR-NB, who completed standard multimodal high-intensity treatment, including high-dose chemotherapy followed by autologous stem cell transplantation and passive immunotherapy with dinutuximab beta, who then receive vaccination with an anti-idiotypic Ab of the murine anti-GD2 mAb 14G2a, will develop a vaccine-specific immune response.

## 2. Materials and Methods

Generation of the ganglidiomab vaccine: The anti-GD2 anti-idiotype Ab (ganglidiomab) was generated as previously described [4]. Briefly, balb/c mice were immunized with monoclonal anti-GD2 Ab 14G2a, and the splenocytes of mice developing anti-14G2a Ab were then fused with SP2/0-Ag 14 non-secreting murine myeloma cells. Hybridomas secreting immunoglobulins bound to the human-mouse chimeric version of 14G2a (ch14.18) were selected and subjected to several rounds of subcloning. Ganglidiomab was identified based on its ability to mimic the nominal antigen GD2. This was tested by blocking the binding of anti-GD2 antibodies of the 14.18 family to GD2 antigen using ELISA [4]. The ganglidiomab vaccine was further characterized structurally and functionally, and its variable (VL and VH) regions were subsequently cloned and sequenced [4]. It was then adsorbed to Alhydrogel^®^ and used at a final concentration of 1 mg/mL. The final product (3.5 mL of a 10 mg/mL, corresponding to 35 mg) was diluted with 27.603 mL of sodium chloride (0.9%) and mixed with 3.8987 mL Alhydrogel^®^ adjuvant 2% (Al(OH)_3_, aluminum hydroxide gel, InvivoGen) (116.91 mg corresponding to 3.34 mg/mL in the final suspension). Aliquots of 0.6 mL of the final suspension were filled into glass vials under sterile conditions and sealed.

Patients: Patients with HR-NB treated at our institution with dinutuximab beta and who achieved complete remission after frontline or salvage therapy were offered the vaccine. All patients or their parents or guardians provided written, informed consent, as appropriate, before compassionate-use treatment was initiated.

Vaccination and assessment of the side effects: Ganglidiomab was administered subcutaneously every two weeks using a dosage of 0.5 mg (0.5 mL) with 1.67 mg Alhydrogel^®^ as an adjuvant. Side effects following the injection of the ganglidiomab vaccine were systematically evaluated on each visit of the patients. The type and severity of side effects were determined according to the Common Terminology Criteria for Adverse Events (CTCAE v.4.03). 

Determination of the humoral immune response: The immune response of immunized patients was determined against three different antigens using enzyme-linked immunosorbent assays (ELISA) (Figure 1).

The first ELISA showed whether the patient developed a response against ganglidiomab (murine IgG1) used for vaccination (Figure 1a) [14]. The second assay analyzed the patient’s response against the variable regions of ganglidiomab (murine Fab), which contains the GD2 mimotope. For this purpose, a human-mouse chimeric version of ganglidiomab (ganglidiximab, of which the Fab region is murine, and the constant regions are human IgG1) was used (Figure 1b) [15]. Finally, the third ELISA analysis showed the patient’s response against ganglioside GD2 (Figure 1c). In each assay, the serum of immunized patients was dispensed into coated microwell plates containing ganglidiomab, ganglidiximab, or GD2 and washed. A horseradish peroxidase (HRP)-enzyme-labeled secondary Ab (goat anti-human Fc Ab or biotinylated ganglidiomab) was then added, and a second wash step removed the unbound secondary antibody. A signal was generated through the addition of 3,3′,5,5′-tetramethylbenzidine (TMB) substrate leading to an enzymatic color change from colorless to blue (through HRP) and eventually to yellow after the addition of sulfuric acid (stop reagent). The signals generated were then measured at 550 nm [14].

Determination of complement-dependent cytotoxicity (CDC) in serum of immunized patients: The GD2-specific CDC of the immunized patients against NB-cells in vitro was determined using a calcein-acetoxymethyl ester (Calcein-AM) release assay [16]. Briefly, LAN-1 cells (human GD2-positive NB-cell line) were labeled with Calcein-AM and used as target cells. Defined concentrations of anti-GD2 mAb dinutuximab beta spiked into the serum of a healthy donor were used as positive controls. As a negative control, the anti-CD20 mAb rituximab was used (LAN-1 cells are CD20 negative). To prove that the observed target cell lysis is mediated by CDC, the humanized mAb eculizumab (trade name Soliris; Alexion Europe SAS, Paris, France), known to selectively inhibit the splitting of complement protein C5 to C5a and C5b by the C5 convertase, was used (data not shown). In patients with a CDC signal, samples were pre-incubated with an excess of the anti-idiotype Ab ganglidiomab to prove GD2 specificity (data not shown).

## 3. Results

### 3.1. Patient Characteristics, Vaccinations and Side Effect Subsection

From March 2013 to November 2014, 7 patients with high-risk NB who completed standard induction, high dose consolidation chemotherapy, and autologous stem cell rescues followed by maintenance therapy and who previously received anti-GD2 immunotherapy with dinutuximab beta at our institution were enrolled in this program (Table 1). The median age at diagnosis was three years (ranging from one month to 25 years). Three of the patients were female, and four patients were male. Four of the patients were *MYCN* amplified. Patient G-03 was initially diagnosed with localized neuroblastoma (stage 2/3) at one month and was negative for *MYCN* amplification. However, in the course of the disease, a metastatic relapse with progression to stage 4 was diagnosed with a concomitant switch to *MYCN* amplification of his neuroblastoma, changing the patient’s status from low-risk to high-risk at the time of relapse. Patients G-02 and G-04 developed relapses after standard high-risk neuroblastoma therapy and achieved second complete responses following salvage therapies. In addition, patient G-02 had a haploidentical stem cell transplantation as part of the second-line therapy regimen. All patients were treated with five cycles of dinutuximab beta as part of their regimen and were in complete remission (CR) before receiving the vaccine (first CR: patients G-01, G-03, G-05, G-06, G-07, second CR: patients G-02 and G-04).

The median interval between initial diagnosis and first vaccine dose and the number of injections are shown (Table 2).

The vaccination was conducted as planned in all patients and consisted of s.c. injections every two weeks. The median interval between the diagnosis and the first vaccine dose was 29 months (range 18–40 months), and the median number of injections was 7 (range 6–22).

### 3.2. Side Effects

Patients were examined, and information was collected about adverse events following the previous vaccination during each visit. The reported adverse events were of low intensity (grade ≤ 2) and short duration (≤48 h). There was no reported severe adverse event; however, all patients disclosed injection-related grade ≤ 2 local reactions that lasted ≤48 h. Fever, pain, neuropathy, or any grade ≥ 3 toxicities did not occur during or post-treatment.

### 3.3. Immune Response

#### 3.3.1. Anti-Ganglidiomab Immune Response

All seven patients received a blood test to determine their baseline values before starting the immunization. After the start of the vaccination, patients were seen in 14-day intervals (up to 168 days). A serum sample was drawn each visit before the next vaccine was given. The first parameter analyzed for all patients refers to the immune response against ganglidiomab (Figure 2).

#### 3.3.2. Anti-Ganglidiximab Immune Response 

The second parameter analyzed is the immune response generated against the chimeric Ab ganglidiximab (Figure 3), which consists of the same murine variable regions as ganglidiomab genetically fused to constant regions of human IgG1 [15]. The same serum samples tested for ganglidiomab were used for ganglidiximab detection.

#### 3.3.3. Anti-GD2 Immune Response 

The immune response generated against the GD2 antigen was detectable in 3 patients with maximum antibody levels reaching 2.5 µg/mL (Figure 4). 

Two patients, G-04 and G-01, showed maximum Ab responses on day 14 and day 84, respectively. The response of patient G-04 was transient with levels falling below 1 µg/mL on day 28. Patient G-03 developed a response from day 14 until the end of the vaccination period but with Ab levels below 1 µg/mL. Again, patient G-02 showed no response, consistent with the non-response observed in the patient’s ganglidiomab and ganglidiximab ELISA methods.

#### 3.3.4. Complement Dependent Cytotoxicity 

The immune response against the ganglidiomab vaccine that translated into a GD2-specific CDC is shown (Figure 5). Similar to the anti-GD2 response, a mixed pattern of CDC activity in individual patients was observed.

Patients G-01 and G-04 developed a CDC response consistent with findings in the GD2, ganglidiomab, and ganglidiximab ELISA methods, suggesting that the vaccine performed as anticipated. The effect observed in these patients was GD2 specific, as indicated by control experiments with the addition of an excess of ganglidiomab (100 µg/mL) in the CDC assay. At all time points with a CDC response of >20%, the addition of ganglidiomab reduced the CDC activity to <5 ± 2%. Interestingly, Patient G-03, who had detectable levels of anti-GD2 antibodies in the serum (Figure 4), did not develop a CDC response (Figure 5). The absence of a CDC response in patients G-05, G-06, and G-07 are consistent with their GD2 ELISA results.

#### 3.3.5. Survival of Immunized Patients 

The date of diagnosis, date of the first vaccine, date of the last follow-up, and patient status are shown (Table 3). The range between the first vaccination and the last follow-up date was determined among the patients enrolled in this program, resulting in a median value of 56 months and 16 days for the entire cohort. Two patients had a relapse or progression of their disease (G-02, and G-04), while the rest were frontline patients. The frontline patients had an overall median range of 56 months and 16 days from their first vaccine dose to their last follow-up date, which is also the range for the entire group. In comparison, the range for the two relapsed patients is 56 months and 20 days for G-04 and 16 months and 19 days for G-02, from their first vaccine dose to their last follow-up date or date of death. No significant side effects were also observed. These observational data suggest that vaccination with ganglidiomab is safe and may be of benefit.

## 4. Discussion

Anti-GD2 antibodies were shown to improve the outcome of children with HR-NB [18], and the introduction of dinutuximab beta for passive immunotherapy of patients with HR-NB also showed an improvement in event-free and overall survival [19].

However, the efficacy of passive immunotherapy is short-lived and decreases once the treatment cycles are completed. Therefore, a vaccine that can provide a sustained anti-tumor response with prolonged protection from relapse may be an important add-on approach for patients who completed passive immunotherapy. For this purpose, the ganglidiomab vaccine was developed and tested in seven patients with HR-NB following passive immunotherapy with dinutuximab beta. The response against the vaccine ganglidiomab was excellent, as well as the response against ganglidiximab containing the GD2 mimotope of the Fab of ganglidiomab (Figure 2 and Figure 3), except for patient G-02.

The response reached a plateau for all six responding patients after six injections when testing against ganglidiomab (murine IgG1) used for vaccination, which corresponds to an effective cumulative vaccine dose of 3 mg ganglidiomab. Patient G-03 received 12 injections (cumulative dose: 6 mg) but did not show a further increase in the response after 6 injections. This suggests that 3 mg ganglidiomab is a useful target dose for a future Phase I clinical trial.

However, the response was slower when testing against the variable region (as represented by the human-mouse chimeric variant ganglidiximab). It is still unclear if a variation in the vaccine dose will result in different response dynamics, which may be subject to a Phase I dose escalation study. 

Patient G-02 who did not respond to the vaccinations had a haploidentical stem cell transplantation as part of the previous treatment. The source for the hematopoietic blood stem cells used for the haploidentical stem cell transplantation of this patient was a leukapheresis product from a parent. T-cells and B-cells were removed from the graft by depleting CD3- (T-cell marker) and CD19- (B-cell marker) positive cells using the CliniMACS technology [20]. This procedure results in a very low incidence of graft versus host disease (GvHD) [21] but is also associated with a prolonged recovery of the immune system with a long-lasting B-cell deficiency [22]. Since the vaccination with ganglidiomab and the subsequent induction of a B-cell response requires both a functional B-and T-cell compartment, the haploidentical transplantation provides a mechanistic explanation for why this patient showed nearly no response following ganglidiomab vaccination. 

The outcome of the patients receiving the vaccine was also encouraging (Table 3), and no clinically significant side effects were seen. No events were observed until the last follow-up except for patient G-02 who died from an anaphylactic reaction during a routine imaging investigation. All patients were in CR before starting the vaccination. Although the sample size of patients in the program is low, the absence of any relapse/progression may constitute a signal of clinical activity. The median observation for the five frontline patients without an event was 56 months and 16 days (>4 years). In a phase 3 multicenter, randomized trial evaluating the effect of dinutuximab beta with IL-2 and dinutuximab beta alone among eligible patients with the primary endpoint being a 3-year event-free survival, EFS was found to be 56% (95% CI 49–63) among patients assigned to dinutuximab beta, compared to 60% (95% CI 53–66) for patients in the dinutuximab beta plus subcutaneous IL-2 group [19]. Since all the patients in our compassionate program received antibody therapy with IL-2, adding ganglidiomab may have potentially increased the EFS and OS of immunized patients. Interestingly, the two patients with relapsed/refractory NB did not experience a relapse during the monitoring phase. For G-02, the time to the last follow-up (referring to the date of death unrelated to neuroblastoma) from the initial diagnosis was 57 months, whereas the time from the first vaccine dose to the last follow-up was 16 months and 19 days. On the other hand, G-04 had a time to follow-up from the date of diagnosis of about 89 months, and the time from the first vaccine dose to the last follow-up was 56 months and 20 days. In contrast, the data derived from a cohort of patients with recurrent/refractory NB from the Children’s Oncology Group (COG) modern-era early phase trials study identified a time-to-progression (TTP) of only 58 days [23]. 

However, a discrepancy was found between the anti-ganglidiomab and anti-ganglidiximab response compared to the anti-GD2 response (Figure 2, Figure 3 and Figure 4) in 3 out of the 7 patients, which may be due to methodological or conceptual reasons. Coating the wells for the ELISA assay with GD2 is a sensitive procedure since glycolipids do not have the same coating properties as proteins. Therefore, wash steps can disrupt the adhesion of the GD2 to the plastic surface of the 96-well plates. One possibility to solve this methodological obstacle may be refinements of the assays and determining the inhibition of the response against ganglidiomab by adding an excess of the antigen GD2 or using a Biacore system [24] with specific biosensors optimized for glycolipids [25]. Furthermore, the analysis of antibody-dependent cellular cytotoxicity (ADCC) and phagocytosis (ADCP), in addition to the CDC response (Figure 5), may provide a more comprehensive assessment of the ganglidiomab vaccination effect. Such assay optimizations are subject to future work. Another conceptual reason for the observed differences in assay response could lie in the choice of the adjuvant. Although aluminum-based adjuvants remain the only adjuvant licensed for human use until 1990 due to their remarkable immunostimulatory properties [26], a growing number of alternatives might be advantageous [27]. These include water-in-oil formulated adjuvants such as MF59 or the incomplete Freund’s adjuvant (IFA) that can enhance Ab responses and are well-tolerated [28]. Another adjuvant available for human use is β-glucan which activates C-type lectin receptors [27] and stimulates the production of granulocytes, monocytes, macrophages, and natural killer cells [29].

Finally, the structure of the antigen used for the vaccine design may also explain the variable results observed. The part of the anti-idiotypic Ab ganglidiomab that mimics the nominal antigen GD2 is a relatively small part of the immunoglobulin molecule, i.e., the complementarity-determining region (CDR). The other parts of the ganglidiomab immunoglobulin molecule are not needed for the desired GD2-specific response. Hence, other vaccine designs, for example, using short peptides that mediate the mimotope function, may provide an advantage. Such GD2 mimotopes of the linear and circular structure were identified by phage display technology [30,31,32] and further optimized by SPOT synthesis [32] (SPOT is a technique of sequential, systematic amino acid substitution). When mimetic peptides are used for immunization, they induce the desired anti-GD2 Ab responses based solely on the principle of molecular mimicry [31,33,34]. A side-by-side comparison of such GD2 peptide mimotopes and ganglidiomab is subject to further investigation. In both examples using the circular and the linear peptide mimotope, a sequential approach of peptide identification followed by optimization was proven successful. This method may therefore provide a strategy to optimize the efficacy of the GD2 peptide mimotope of ganglidiomab. In another Phase II Trial using the gangliosides, GD2L/GD3L as a vaccine with oral β-glucan as an adjuvant, anti-GD2 IgG1 Ab (titers ≥ 150mg/L) was found to be associated with improving PFS and OS (32 ± 6% and 71 ± 7% at five years, respectively) of patients, with serum titers increasing at week six following the administration of β-glucan. In addition, the SNP rs3901533 was also shown to be a biomarker of the antibody response. However, this study also showed that anti-GD3-IgG1 and anti-GD2-IgM were not significantly improving survival [13]. Data from the above studies further emphasize the role of active immunity in controlling NB. The consistent presence of GD2 Ab in the tumor milieu, as opposed to regular injections of mAb, could potentially improve tumor control.

## 5. Conclusions

We report the development of an anti-idiotype vaccine (ganglidiomab) against the tumor-associated antigen disialoganglioside GD2 and describe the immune response in seven vaccinated patients. 

None of the immunized patients experienced a relapse of their NB, with a median range of 56 months and 16 days from their first dose of the vaccine to their last follow-up, contrasting with what is known from historical control cohorts. This is the first-in-man use of the anti-idiotype vaccine ganglidiomab providing important baseline data to evaluate the vaccine in prospective clinical trials.

## Figures and Tables

**Figure 1 cancers-14-05802-f001:**
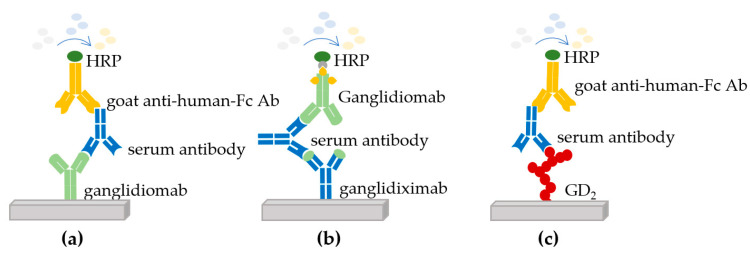
Enzyme-linked immunosorbent assays (ELISA) used to determine the immune response of immunized patients. Three different antigens were used to perform ELISAs to determine the response in the serum of immunized patients: (**a**) against the ganglidiomab vaccine, (**b**) against the human-mouse chimeric antibody ganglidiximab, and (**c**) against GD_2_.

**Figure 2 cancers-14-05802-f002:**
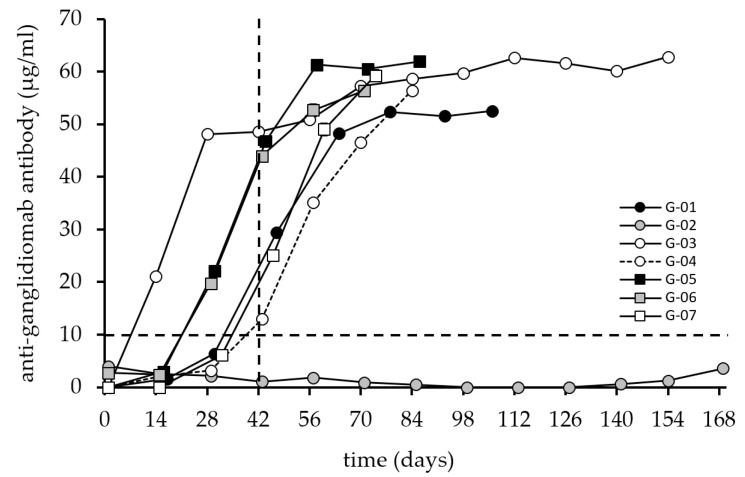
Induction of anti-ganglidiomab humoral response after vaccination with ganglidiomab over time. The anti-ganglidiomab serum levels were analyzed by ELISA following the procedure described in the materials and methods. Patients received 0.5 mL of 1 mg/mL of ganglidiomab adsorbed to Alhydrogel^®^ in 14-day intervals. The patients were immunized 6–22 times with ganglidiomab every two weeks, and blood samples were collected on vaccination days at the indicated time points. The graphs indicate the mean value for each patient at each visit. The standard deviation (error bar) is too small to display and is covered by the size of the symbol. The dashed lines indicate the 10 µg/mL value (horizontal) and the 42-day time point after three vaccinations (vertical) for illustration purposes.

**Figure 3 cancers-14-05802-f003:**
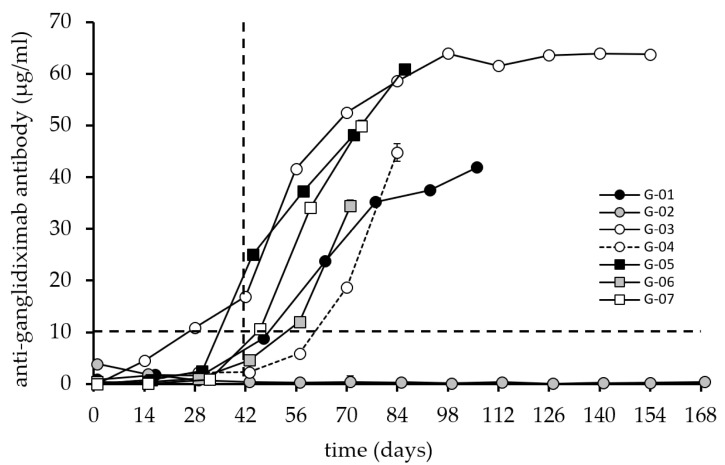
Induction of an anti-ganglidiximab humoral response after vaccination with ganglidiomab. The anti-ganglidiximab response serum levels were analyzed by ELISA following the procedure described previously. The graphs indicate the mean value for each patient at each visit. The standard deviation (error bar) is too small to display and is covered by the size of the symbol. The dashed lines indicate the 10 µg/mL level (horizontal) and the 42-day time point after three vaccinations (vertical) for illustration purposes.

**Figure 4 cancers-14-05802-f004:**
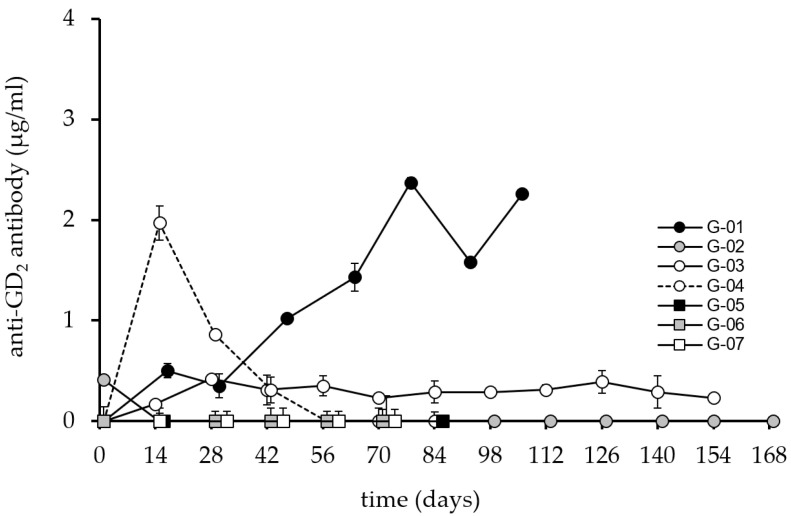
Induction of anti-GD2 humoral response after vaccination with ganglidiomab over time. The anti- GD2 response serum levels were analyzed by ELISA following the procedure described previously for the seven patients immunized with the ganglidiomab vaccine. The graphs indicate the mean value for each patient at each visit. The standard deviation (error bar) is small and can only be seen in some of the values presented in the graph.

**Figure 5 cancers-14-05802-f005:**
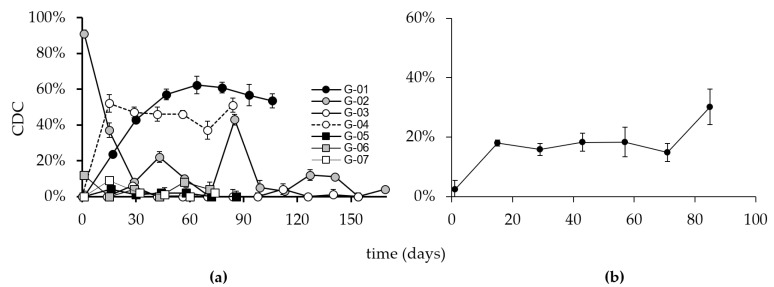
CDC activity in the sera of immunized patients. Blood samples are collected on the indicated days, as shown in the above figure. Sera of the seven patients were then analyzed using calcein-AM-based cytotoxicity assay as described in the methods. (**a**) Percent activity is depicted in the above figure for each patient. (**b**) Mean CDC activity of all patients. *Y*-axis: CDC% of target cell lysis, *X*-axis: vaccination time points. Error bars indicate the standard deviation.

**Table 1 cancers-14-05802-t001:** Characteristics of patients enrolled in the compassionate-use treatment program.

Patient	Age at Diagnosis	Sex	Relapse	BSA ^1^	Stage ^2^	*MYCN* Amplification
G-01	3 years	M	No	0.9 m^2^	4	no
G-02	25 years	F	Yes	1.5 m^2^	4	no
G-03	1 month	M	No	0.6 m^2^	4	yes
G-04	3 years	F	Yes	0.6 m^2^	4	unknown
G-05	6 months	M	No	0.5 m^2^	4	yes
G-06	7 years	F	No	1.1 m^2^	4	yes
G-07	5 years	M	No	0.8 m^2^	4	yes

^1^ BSA = body surface area, ^2^ Stage according to the international neuroblastoma staging system (INSS) [17]. M, male; F, female.

**Table 2 cancers-14-05802-t002:** Interval between diagnosis and first vaccine and total number of vaccinations.

Patient	Time Interval Diagnosis-First Dose	Number of Doses Received
G-01	29 months: 10/2010–11/2012	8
G-02	40 months: 01/2010–05/2013	22
G-03	34 months: 05/2011–03/2014	12
G-04	33 months: 02/2012–11/2014	7
G-05	22 months: 01/2013–11/2014	7
G-06	18 months: 05/2013–11/2014	6
G-07	29 months: 06/2012–11/2014	6

**Table 3 cancers-14-05802-t003:** Patient status and range of the first vaccine dose to the last follow-up.

Patient	Date of Diagnosis	Date of First Vaccination	Date of Last Vaccination	Date of Last Follow-Up	Range from First Dose to Last Follow-Up	Patient Status
G-01	10/2010	20/11/2012	07/03/2013	26/07/2019	80 mo, 7 d	NED
G-02 ^1^	01/2010	22/05/2013	12/03/2014	10/10/2014 ^2^	16 mo, 19 d	Dead ^3^
G-03	05/2011	25/03/2014	26/08/2014	28/06/2019	63 mo, 4 d	NED
G-04 ^1^	02/2012	12/11/2014	07/01/2015	31/07/2019	56 mo, 20 d	NED
G-05	01/2013	03/11/2014	21/01/2015	18/11/2018	48 mo, 16 d	NED
G-06	05/2013	17/11/2014	26/01/2015	01/08/2019	56 mo, 16 d	NED
G-07	06/2012	14/11/2014	09/02/2015	01/07/2015	7 mo, 18 d	NED

^1^ Patients with relapsed high-risk neuroblastoma, ^2^ Date of last follow-up refers to the patient’s date of death, which was unrelated to neuroblastoma, ^3^ Death not related to neuroblastoma, months; d, days; NED, alive and no evidence of disease.

## Data Availability

The data presented in this study are available in this article.

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
