# Peer review of "Immune Response and Outcome of High-Risk Neuroblastoma Patients Immunized with Anti-Idiotypic Antibody Ganglidiomab: Results from Compassionate-Use Treatments"

_cancers, 2022, doi:10.3390/cancers14235802_

Round 1

Reviewer 1 Report (Previous Reviewer 1)

Thanks for the answers.

Author Response

Thank you for accepting the manuscript, we have improved it according to your suggestions.

Reviewer 2 Report (Previous Reviewer 2)

In this study, Leah Klingel and co-workers describe the first-in-man compassionate use of ganglidiomab as a vaccine following passive immunotherapy with anti-GD2 antibody (dinutuximab beta) in seven patients. The vaccine was well tolerated, and all patients not pre-treated by haploidentical transplantation developed vaccine-specific immune responses. These findings provide an important basis for the design of prospective clinical trials. Although this study is of good quality, to my opinion, several points need to be addressed before the article may be formally accepted for publication:

1)The amount, quantity, and frequency of antigens entering the body can significantly affect the strength and type of the body's immune response to the antigen. “5 frontline, 2 relapsed”: Throughout the article, whether different types of patients need to be discussed separately.

2)A segmented discussion is recommended in the Discussion section of the article, and it is necessary to highlight the interpretation of the patient "pre-treated by haploidentical transplantation".

3)The doses of vaccines received by different patients are different, so please make a reasonable explanation for the impact of the dose on the final experimental results.

4)It is reasonable to test the amount of variation of "anti-GD2 Ab", but why track tests for two other antibodies such as "Anti-ganglidiximab antibody" and "Anti-ganglidiximab antibody"

5)“HR-NBL” on line 260: Abbreviations throughout the article suggest specification and uniformity.

6)The English form is sometimes poor and needs rigorous revision before re-submission.

7)Why do some points have bar values in Figure 3 and Figure 4, but some points do not have corresponding bar values?

Considering the scientific value of the work, I do recommend its publication in Cancers after a further major revision of its form and layout.

Author Response

1)The amount, quantity, and frequency of antigens entering the body can significantly affect the strength and type of the body's immune response to the antigen. “5 frontline, 2 relapsed”: Throughout the article, whether different types of patients need to be discussed separately.

We agree with the reviewer that quantity and frequency of vaccination with the antigen ganglidiomab may affect the dynamics of immune response to the antigen. In fact, we show a frequency dependent induction of a humoral response against the vaccine (Figures 2 and 3).

The response reaches a plateau for all six responding patients after the 6th injection when testing against murine IgG ganglidiomab used for vaccination. However, the response was slower when testing against the variable region only (as represented by the human/mouse chimeric variant ganglidiximab) and this was described in section 3.3. We did not evaluate the quantity in this small series of patients, but we agree that to increase the injected dose from 0,5 mg to 1,0 mg, or to decrease it from 0,5 mg to 0,1 mg or even lover may affect the kinetics of the response as well. However, the intention of this first in man use study was to determine the feasibility and the response as a first data point, which will help now to design a phase I study to address the questions raised by the reviewer. We added three sentences for clarification to the discussion (lines 279-287).

Related to status of the patients in this series, they were all high-risk neuroblastoma patients who completed standard induction, consolidation and maintenance therapy. This was clarified in the revised version in line 149-150. and 5 of them received the vaccination subsequent to maintenance therapy with dinutuximab beta. Two patients experienced a relapse and completed second line regiment to the point where they achieved a second CR before receiving the vaccine. We added more details related to the patient characteristics to section 3.1.

We are aware of the heterogeneity of the high-risk neuroblastoma patient population which is described in section 3.1. However, the common denominators are the complete response and the treatment with dinutuximab beta. The patient receiving the haplo transplant was discussed in detail as we think that the B-cell deficiency may explain the absence of a response in this patient. However, we do not think that a patient by patient discussion will be helpful to the reader.

2)A segmented discussion is recommended in the Discussion section of the article, and it is necessary to highlight the interpretation of the patient "pre-treated by haploidentical transplantation".

We segmented the discussion according to the following topics each starting with a new paragraph:

  • Patient G-02 who had a haploidentical stem cell transplantation
  • Outcome results
  • Assay discussion
  • Vaccine design

We think that this type of format change is helpful to the reader, thank you for the comment.

3)The doses of vaccines received by different patients are different, so please make a reasonable explanation for the impact of the dose on the final experimental results.

Thank you for that comment. The number of doses given to patient G-02 was 2-3 time higher compared to the other patients since the expectation was to eventually induce a response at a later time point by continuing with vaccinations and thereby increasing the cumulative vaccine dose. This did not translate into the expected effect, most likely due to the persistent B-cell deficiency in that haplo-transplanted patient.

Five patients had 6-8 injections, which could be clustered together as being a similar dose range, and the responses observed followed a similar kinetics, reaching a plateau after 6 injections (=3 mg) (Figure 2).

Patient G-03 had 12 injections. The response of this patient also reached a plateau after 6 injections with no further increase, suggesting that 6 injections might be target dose for future trials. We added a statement related to this observation to the discussion (line 280-284). 

4)It is reasonable to test the amount of variation of "anti-GD2 Ab", but why track tests for two other antibodies such as "Anti-ganglidiximab antibody" and "Anti-ganglidiximab antibody"

The stepwise testing strategy is necessary at the moment since there are methodological challenges with the anti-GD2 assay as discussed in detail in the discussion section (line 322-332). The GD2 mimotope is a structural part of the variable region of ganglidiomab, therefore the human-mouse chimeric variant of the vaccine is a helpful additional test. However, once the GD2 assay is optimized the focus will be directed to that readout. No changes made to the manuscript.

5)“HR-NBL” on line 260: Abbreviations throughout the article suggest specification and uniformity.

HR-NBL was a mistake, it should read HR-NB, and is now corrected. This abbreviation for high-risk neuroblastoma was defined in the abstract.

6)The English form is sometimes poor and needs rigorous revision before re-submission.

We did a rigorous revision, and hope it is now sufficient to proceed with the manuscript.

7)Why do some points have bar values in Figure 3 and Figure 4, but some points do not have corresponding bar values?

At the points where bar values are not visible, they are covered by the symbol. Respective statements are added to the respective figure legends. 

Considering the scientific value of the work, I do recommend its publication in Cancers after a further major revision of its form and layout.

Thank you for this assessment, and please find the major revision uploaded.

Reviewer 3 Report (New Reviewer)

The article report a compassionate use of ganglidiomab as a vaccine following immunotherapy with dinutuximab beta.

They reported data from a limited number of patients only 7 but  the data of response and long term follow-up are interesting and encouraging.

Side effects need a better description, not clear " common toxicities were self-limiting"

Materials and methods section is not sufficiet detailed

Author Response

The article report a compassionate use of ganglidiomab as a vaccine following immunotherapy with dinutuximab beta.

They reported data from a limited number of patients only 7 but  the data of response and long term follow-up are interesting and encouraging.

Thank you very much for this assessment.

Side effects need a better description, not clear " common toxicities were self-limiting"

We added more detail to the description of adverse events in the revised version.

Materials and methods section is not sufficient detailed

Details of the materials and methods section are in part reported in other manuscripts, which were not always referenced where necessary. This was corrected in the revised version.  

Round 2

Reviewer 2 Report (Previous Reviewer 2)

The authors have addressed my concern, now it can be published. 

This manuscript is a resubmission of an earlier submission. The following is a list of the peer review reports and author responses from that submission.

Round 1

Reviewer 1 Report

Authors present a very limited first ever clinical experience using ganglidiomab (anti-idiotype antibody) as vaccine in 7 patients entering complete remission after multimodality treatment including anti-GD2 immunotherapy with dinutuximab beta. They present serum data showing specific immunoreactivity against ganglidiomab (anti-idiotype Ab), ganglidiximab (chimeric Ab) but not against GD2. The vaccination occurred under compassionate use close to 10 years ago (2013-14). Authors claim an improved outcome of these patients.

Some relevant issues:

1. Compassionate use is not an ethically correct form to test first in human products. First in human studies require a clinical trial format to evaluate appropriately the quality of a new product in humans. Evaluation by local ethical committee should be required to present first in human data.

2. Patients were vaccinated from 2013 and 2014 when ganglidiomab was first reported by the same group. Why now almost 10 years ago are these patients reported? why clinical development has not occurred for ganglidiomab?

3. The 7 patients selected are not representative of the high-risk neuroblastoma patient spectrum. 4 out of the 7 are MYCN amplified (>50%) a clear bias towards a subtype of neuroblastoma which is characterized by a high rate of cure (high EFS and OS) if patients achieve complete remission, like those presented.

4. Outcome data for 7 patients who achieved CR is irrelevant mainly when majority are MYCN amplified. All comparisons with published data for large cohorts of high-risk neuroblastoma patients is speculative.

5. References in parts of the introduction section are not adequate. Treatment with therapeutic MIBG is not part of standard management (ref 2 is inadequate). Refewrences 3 and 4 are not adequate for current advances (both are form 2013). Reference 7 is not adequate to discuss prognosis of patients. 

6. Typo: 10e7 molecules per NB cell is what is reported in ref 8 (not 107)

4.  

Author Response

We provide a point by point rebuttal in blue and italic font.

Authors present a very limited first ever clinical experience using ganglidiomab (anti-idiotype antibody) as vaccine in 7 patients entering complete remission after multimodality treatment including anti-GD2 immunotherapy with dinutuximab beta. They present serum data showing specific immunoreactivity against ganglidiomab (anti-idiotype Ab), ganglidiximab (chimeric Ab) but not against GD2. The vaccination occurred under compassionate use close to 10 years ago (2013-14). Authors claim an improved outcome of these patients.

We would like to clarify that we found a GD2 response in 3 of the 7 vaccinated patients. Explanations of discrepancies are provided in a detailed discussion with adaptations for future work.

Some relevant issues:

  1. Compassionate use is not an ethically correct form to test first in human products. First in human studies require a clinical trial format to evaluate appropriately the quality of a new product in humans. Evaluation by local ethical committee should be required to present first in human data.

We confirm that compassionate use in Germany as indicated under “individueller Heilversuch” does not require authorisation or evaluation by ethical committees or competent authorities (“Individuelle Heilversuche sind im Rahmen der ärztlichen Therapiefreiheit grundsätzlich zulässig; es besteht keine Genehmigungs- oder Anzeigepflicht”). It is clearly stated in the manuscript that the next step will be a clinical trial following a formal approval process.

  1. Patients were vaccinated from 2013 and 2014 when ganglidiomab was first reported by the same group. Why now almost 10 years ago are these patients reported? why clinical development has not occurred for ganglidiomab?

The main reason for the time span this related to valuable follow up information collected during that period. The rate limiting step of the ongoing clinical development process for this product is monetary resources available for the development in a competitive segment of an orphan disease with drugs such as DFMO, naxitamab, dinutuximab and dinutuximab beta, just to name a few. However, we are optimistic to continue our efforts to advance with this project.

  1. The 7 patients selected are not representative of the high-risk neuroblastoma patient spectrum. 4 out of the 7 are MYCN amplified (>50%) a clear bias towards a subtype of neuroblastoma which is characterized by a high rate of cure (high EFS and OS) if patients achieve complete remission, like those presented.
  2. Outcome data for 7 patients who achieved CR is irrelevant mainly when majority are MYCN amplified. All comparisons with published data for large cohorts of high-risk neuroblastoma patients is speculative.

The following paragraph constitutes a response to the reviewers points 3 and 4. The usual representation of MYCN amplification at initial diagnosis is around 30% in patients with metastatic high risk neuroblastoma. With n=7 patients treated with ganglidiomab, it not possible to appropriately balance the patient population for MYCN amplification, which would be subject to future clinical trials. We are also careful related to statements about our outcome observations reported here, given the fact that the patient number is low. However, it is fair to indicate that the absence of an event supports an interest to continue clinical research with this approach.

  1. References in parts of the introduction section are not adequate. Treatment with therapeutic MIBG is not part of standard management (ref 2 is inadequate). Refewrences 3 and 4 are not adequate for current advances (both are form 2013). Reference 7 is not adequate to discuss prognosis of patients.

We partially agree with the reviewer’s suggestion related to accuracy of reference 2, and therefore exchanged the reference with the current guideline for the treatment of neuroblastoma of the German Pediatric Hematology and Oncology group (https://pubmed.ncbi.nlm.nih.gov/28561228/), which is standard for the centres participating to results reported here. We kept the statement related to mIBG therapy as this is a recommendation in that guideline in particular for patients with mIBG avid disease, although it may not be standard for other countries or cooperative groups working in the field.

The references 3 and 4 were used to indicated advancements in the treatment of neuroblastoma. We agree with the reviewer that there could be a more suitable reference, and we therefore exchanged reference 3 with a review from Suzanne MacFarland and Rochelle Bagatell entitled “Advances in neuroblastoma therapy” (https://pubmed.ncbi.nlm.nih.gov/30480556/), although the older review from N. Chung is more focussed on immunotherapy which is after all the primary topic of this report. Reference 4 was kept as this reference reflects important previous work relevant to this manuscript.

Reference 7 was exchanged with a review of Pinto NR et al on “Advances in Risk Classification and Treatment Strategies for Neuroblastoma” which indicates the development of probability of overall survival among 3,352 high risk neuroblastoma patients according to treatment era (https://pubmed.ncbi.nlm.nih.gov/26304901/).

  1. Typo: 10e7 molecules per NB cell is what is reported in ref 8 (not 107)

Typo was corrected.

Reviewer 2 Report

Neuroblastoma is a type of rare cancer that develops in extracranial nerve tissue and commonly occurs in children. However, current therapies tend to associate with dismal prognosis. Therefore, the development of more effective therapeutic strategies for advanced neuroblastoma is of great clinical significance. In this work, the authors evaluated the effect of ganglidiomab-based anti-idiotypic vaccine in compassionate use treatments of seven patients. The study is very interesting and meaningful, the research was also well conducted. Nevertheless, there is still one issue need to be further discussed:

The complement dependent cytotoxicity was determined in order to evaluate the immune response of the vaccine from the aspect of mechanism. However, the results were not completely consistent with that of the anti-GD2 immune response of all patients. ADCC and ADCP are also the main mechanisms of the function of mAbs, should these two factors be considered and determined so that the immune response can be evaluated accurately and comprehensively?

Otherwise, this manuscript is suggested for its acceptance.

Author Response

Thank you very much for your advice regarding an expansion of the immunomonitoring of patients treated with the ganglidiomab vaccine. In the present evaluation we focussed the investigations on the CDC response for practical reasons, since only serum samples are needed to perform the assay. ADCC and ADCP require isolation of cells and subsequent ADCC and ADCP assays. We are in fact currently working to establish sensitive methodology in this direction using Incucyte® live cell analysis systems (https://intellicyt.com/products/live-cell-analysis-system/) that allow the evaluation of immune responses over several days. The plan is to implement this technology in a prospective clinical trial.    

We added a sentence in the discussion part indicating this direction of future research (line 327-329)

Reviewer 3 Report

The authors provide an interesting analysis of the immune response and outcome of patients with high-risk neuroblastoma immunised with the anti-idiotype antibody ganglidiomab. They report data from a limited number of patients (7), 6 of whom were effectively immunised and had no relapse of their NB with a median interval. Although the study is undoubtedly preliminary, it is riddled with numerous limitations of which the authors seem to be aware, so much so that they discuss them at length. 

Main concerns

Although the data reported are encouraging, especially considering the near absence of side effects of the vaccination approach, there is insufficient evidence that the observed effect is mediated by immunisation against the GD2 antigen. In fact, as noted by the authors themselves, there was a discrepancy between the anti-ganglidiomab and anti-ganglidiximab response compared to the anti-GD2 response (Figures 2-4) in 3 of the 7 patients.  In this regard: 

1) The authors should include in the text the controls they refer to on page 4. lines 146-148: "In patients with a CDC signal, samples were pre-incubated with an excess of anti-idiotype Ab ganglidiomab to also demonstrate GD2 specificity (data not shown)".

2) To partially solve the problem of the discrepancy found in the ELISA, the authors could perform a competition assay by running the assay as described in Figure 1A and adding increasing doses of GD2 to test its ability to interfere with the binding between serum antibodies and ganglidiomab.

3) Should the authors have access to the cellular component of the blood samples of the patients included in the study, it would be interesting to see if they have a subset of memory B-cells capable of being reactivated and differentiating into plasma cells upon stimulation with the disialoganglioside GD2. 

Minor problems 

The materials and methods section is not sufficiently detailed. The authors should more clearly state the methods and reagents used to obtain the results described, including the procedures for collecting and processing blood samples. 

Author Response

We provide a point by point response to the reviewers criticisms in italics and blue.

The authors provide an interesting analysis of the immune response and outcome of patients with high-risk neuroblastoma immunised with the anti-idiotype antibody ganglidiomab. They report data from a limited number of patients (7), 6 of whom were effectively immunised and had no relapse of their NB with a median interval. Although the study is undoubtedly preliminary, it is riddled with numerous limitations of which the authors seem to be aware, so much so that they discuss them at length.

We confirm the assessment of the reviewer that the encouraging clinical observations triggered a report of these patients, which we consider an important addition to current developments in the space of immunotherapy for neuroblastoma. We also share the reviewer’s assessment that the nature of the report encompasses a critical evaluation and discussion of the effects observed in concomitant investigations also in light of a first evaluation of this approach in patients.

Main concerns

Although the data reported are encouraging, especially considering the near absence of side effects of the vaccination approach, there is insufficient evidence that the observed effect is mediated by immunisation against the GD2 antigen. In fact, as noted by the authors themselves, there was a discrepancy between the anti-ganglidiomab and anti-ganglidiximab response compared to the anti-GD2 response (Figures 2-4) in 3 of the 7 patients.  In this regard:

1) The authors should include in the text the controls they refer to on page 4. lines 146-148: "In patients with a CDC signal, samples were pre-incubated with an excess of anti-idiotype Ab ganglidiomab to also demonstrate GD2 specificity (data not shown)".

Thank you very much for this comment, we followed the suggestion in the revised version of the manuscript, and added the data to the text (lines 246-248). This addition now clarifies the GD2 specificity of the observed responses in patients 01 and 04. The possible explanations for the discrepancies are detailed in a critical discussion.

2) To partially solve the problem of the discrepancy found in the ELISA, the authors could perform a competition assay by running the assay as described in Figure 1A and adding increasing doses of GD2 to test its ability to interfere with the binding between serum antibodies and ganglidiomab.

This is a very interesting idea, thank you very much for the suggestion. As detailed in the discussion, we are aiming at improving the bioassays used to determined the immune response observed in vaccinated patients. We added a sentence in the discussion to reflect the use of excess antigen to also look at the inhibition of the response against the vaccine (line 324-325).

3) Should the authors have access to the cellular component of the blood samples of the patients included in the study, it would be interesting to see if they have a subset of memory B-cells capable of being reactivated and differentiating into plasma cells upon stimulation with the disialoganglioside GD2.

This is a helpful suggestion for future research. It can not be realized easily as no cellular material was stored from the patients, but will be considered for upcoming activities.

Minor problems

The materials and methods section is not sufficiently detailed. The authors should more clearly state the methods and reagents used to obtain the results described, including the procedures for collecting and processing blood samples.

We have previously reported on the assay development used for the immune monitoring of the vaccinated patients in detail, and all the manuscripts were referenced:

  1. Siebert, N., et al., Validated detection of human anti-chimeric immune responses in serum of neuroblastoma patients treated with ch14.18/CHO. J Immunol Methods, 2014. 407: p. 108-15.
  2. Eger, C., et al., Generation and Characterization of a Human/Mouse Chimeric GD2-Mimicking Anti-Idiotype Antibody Ganglidiximab for Active Immunotherapy against Neuroblastoma. PLoS One, 2016. 11(3): p. e0150479.
  3. Siebert, N., et al., Validated detection of human anti-chimeric immune responses in serum of neuroblastoma patients treated with ch14.18/CHO. J. Immunol. Methods, 2014. 407: p. 108-115.
  4. Siebert, N., et al., Functional bioassays for immune monitoring of high-risk neuroblastoma patients treated with ch14.18/CHO anti-GD2 antibody. PLoS One, 2014. 9(9): p. e107692.

They also contain detailed information about blood sampling, processing and storage.